# Fast exact recovery of noisy matrix from few entries: the infinity norm approach

**BaoLinh Tran and Van Vu**
Department of Mathematics
Yale University
New Haven, CT 06511
l.tran@yale.edu and van.vu@yale.edu

## Abstract

The matrix recovery (completion) problem, a central problem in data science, involves recovering a matrix $A$ from a relatively small random set of entries. While such a task is generally impossible, it has been shown that one can recover $A$ exactly in polynomial time, with high probability, under three basic and necessary assumptions: (1) the rank of $A$ is very small compared to its dimensions (low rank), (2) $A$ has delocalized singular vectors (incoherence), and (3) the sample size is sufficiently large. Various algorithms address this task, including convex optimization by Candes, Recht, and Tao (2009), alternating projection by Hardt and Wooters (2014), and low-rank approximation with gradient descent by Keshavan, Montanari, and Oh (2009, 2010). In applications, Candes and Plan (2009) noted that it is more realistic to assume noisy observations. In such cases, the above approaches provide approximate recovery with small root mean square error, which is difficult to convert into exact recovery. Recently, results by Abbe et al. (2017) and Bhardwaj et al. (2023) on approximation in the infinity norm showed that one can recover $A$ even in the noisy case, provided $A$ has bounded precision. However, beyond the three basic assumptions, they either required that the condition number of $A$ be small (Abbe and Fan, 2017) or that the gaps between consecutive singular values be large (Bhardwaj et al., 2023). These additional assumptions conflict, with one requiring singular values to be close together and the other suggesting they should be far apart. It is thus natural to conjecture that neither is necessary. In this paper, we demonstrate that this is indeed the case. We propose a simple algorithm for exact recovery of noisy data, relying solely on the three basic assumptions. The core step of the algorithm is a straightforward truncated singular value decomposition, which is highly efficient. To analyze the algorithm, we prove a new infinity norm version of the classical Davis-Kahan perturbation theorem, improving an earlier result in (Bhardwaj et al., 2023). Our proof employs a combinatorial contour integration argument and is entirely distinct from all previous approaches.

## 1 Introduction

### 1.1 Problem description

A large matrix $A \in \mathbb{R}^{m \times n}$ is hidden, except for a few revealed entries in a set $\Omega \subset [m] \times [n]$. We call $\Omega$ the set of *observations* or *samples*. The matrix $A_\Omega$, defined by

$$(A_\Omega)_{ij} = A_{ij} \text{ for } (i,j) \in \Omega, \text{ and } 0 \text{ otherwise,} \tag{1}$$

is called the *observed* or *sample* matrix. The task is to recover $A$, given $A_\Omega$. This is the *matrix recovery* (or *matrix completion*) problem, a central problem in data science that has received significant

39th Conference on Neural Information Processing Systems (NeurIPS 2025).

attention in recent years, motivated by a number of real-world applications. Examples include building recommendation systems, notably the **Netflix challenge** [1]; reconstructing a low-dimensional surface based on partial distance measurements from a sparse network of sensors [2, 3]; repairing missing pixels in images [4]; and system identification in control [5]. See the surveys by Li et al. [6] and Davenport and Romberg [7] for additional applications.

Researchers have proposed two models: (a) $\Omega$ is sampled uniformly among subsets with the same size, or (b) $\Omega$ has independently chosen entries, each with the same probability $p$, called the *sampling density*, which can be known or hidden. The models are interchangeable in mathematical analysis through a simple conditioning trick [4]. Most papers use (b), and we will do so in our paper. Very recently, researchers have also explored models where the sample entries are correlated [8–11]; these are beyond the scope of this paper.

In this paper, we focus on exact recovery (to find all entries exactly). For this problem to make sense, it is important to assume that $A$ has a *fixed precision*, namely all entries of $A$ are integer multiples of a positive constant $\varepsilon$. (Otherwise, it is impossible even to write down an entry exactly, let alone compute it.) In most practical contexts, $\varepsilon$ does not depend on the size of the matrix. For instance, in the Netflix problem, all entries are half-integers, so $\varepsilon = 1/2$. If all entries have two decimal places, then $\varepsilon = 0.01$.

It has been pointed out by Candes and Plan [12] that in practice, data is often perturbed by noise, so we can only observe a partially hidden and noisy version of $A$. The main goal of this paper is to find an exact recovery for $A$ from such a noisy sample.

## 1.2 Basic notation and assumptions

Throughout this paper, $A$ is an $m \times n$ matrix, with $N := \max\{m, n\}$. Consider the SVD of $A = U\Sigma V^T = \sum_{i=1}^{r} \sigma_i u_i v_i^T$, where $r := \operatorname{rank} A$, and the singular values are ordered: $\sigma_1 \geq \sigma_2 \geq \cdots \geq \sigma_r$. We write $A_s = \sum_{i=1}^{s} \sigma_i u_i v_i^T$ for the best rank-$s$ approximation of $A$. An important parameter that appears in many papers in this area is the *condition number* of $A$: $\kappa = \kappa(A) := \sigma_1/\sigma_r$.

The *coherence parameter* is given by $\mu_0 = \mu_0(A) = \max\{\mu(U), \mu(V)\}$, where

$$\mu(U) := \max_{i \in [m]} \frac{m}{r} \|e_i^T U\|^2 = \frac{m\|U\|_{2,\infty}^2}{r}, \tag{2}$$

and analogously for $\mu(V)$. The 2-to-$\infty$ norm of a matrix $M$ is given by $\|M\|_{2,\infty} := \sup\{\|Mu\|_\infty : \|u\|_2 = 1\}$. It should be noted that $\|M\|_{2,\infty}$ is simply the largest L2 norm among the rows of $M$.

We use $C$ to denote a positive constant, whose value depends on the context. When $C$ depends on a set of parameters $a_1, a_2, \ldots, a_k$, we write $C(a_1, a_2, \ldots, a_k)$.

In most existing works on matrix recovery, researchers make the following three assumptions

- *Low-rank:* One assumes that $r := \operatorname{rank} A$ is much smaller than $\min\{m, n\}$. Many papers assume $r$ is bounded ($r = O(1)$), while $m, n \to \infty$.

- *Incoherence:* One requires that the rows and columns of $A$ are sufficiently "spread out", so the information does not concentrate in a small set of entries, which could be easily overlooked by random sampling. In technical terms, one needs $\mu_0$ to be small.

- *Sufficient sampling size/density:* The sampling density (or the size of $\Omega$) is sufficiently large. To ensure all rows and columns are sampled, one needs $|\Omega| \geq CN \log N$. We will typically work in the regime $|\Omega| = N \log^{O(1)} N$, which is optimal up to a logarithmic term.

These assumptions have been shown to be necessary; see [13, 4, 7], and have been used in most papers on exact recovery. We will refer to them as the **basic assumptions**.

In the noisy setting, we observe entries from $A + Z$, a noisy version of $A$, where $Z$ represents the noise matrix. Most studies assume that $Z$ has independent entries with mean 0, but not necessarily from the same distributions. In many papers directly related to our work, it was assumed that $Z$ has bounded entries with probability 1. We will do the same here but note that we can replace this with weaker conditions, such as the entries of $Z$ having light tails (via a standard truncation trick). We let $A_{\Omega,Z} := (A + Z)_\Omega$ denote the noisy observed matrix.

### 1.3 A brief summary of existing results and our contributions

There is a vast literature on matrix completion. The problem of exact recovery in the pure (noiseless) case, under the three basic assumptions, was first solved by Candès and Recht [13], Candès and Tao [4], Recht [14], using *nuclear norm minimization* (**NNM**), which seeks to find the matrix with the smallest nuclear norm that satisfies the observed samples via *semidefinite programming* (**SDP**). Their idea is based on convexifying the intuitive but NP-hard approach of minimizing the rank given the observations. However, the best-known solvers for the SDP run in time $O(|\Omega|^2 N^2)$, which is $O(N^4 \log^2 N)$ in the best case [6], and the calculation may be sensitive to noise [4]. Therefore, it is reasonable to seek faster and more noise-robust algorithms, potentially sacrificing some generality.

In Hardt and Wootters [15], Hardt [16], the authors offered another intuitive but NP-hard approach: minimize $\|X_\Omega - A_\Omega\|_F$ subject to $\operatorname{rank} A = r$, and proposed an approximate solution with *alternating projections*, which switches between optimizing the column and row spaces, given the other. In Keshavan et al. [17, 18], the authors proposed taking a low-rank approximation of $A_\Omega$ via SVD, then using it as an initial value for a GD-based algorithm to approximate $A$. In both approaches, it is important that the condition number $\kappa$ be small. See [19, Section A] in the supplementary appendix for a more detailed discussion of the role of $\kappa$ in these results. See also [20–23].

Recovery with noisy data, while more practical, as pointed out by Candes and Plan [12] in their influential survey, is clearly a harder problem. The authors of the methods discussed above all extended their studies to this case, achieving approximate solutions with guarantees in the Frobenius norm. However, it seems very difficult to turn these approximations into exact recovery.

Recently, there has been progress in achieving *exact recovery* in the noisy case by Abbe and Fan [24] and Bhardwaj et al. [25], with simple and fast algorithms. The basic idea is to show that a properly chosen low-rank approximation of the observed (noisy) matrix, under an appropriate assumption, approximates the ground truth matrix $A$ in the infinity norm, with an error less than $\varepsilon/2$, where $\varepsilon$ is the discretization unit of the entries of $A$. Once this is achieved, a simple rounding off recovers $A$ exactly (see the last few paragraphs of Subsection 1.1). The heart of the matter is the analysis of the algorithms, which requires new mathematical ideas, as proving approximation in the infinity norm is significantly more challenging than a similar task under the Frobenius or spectral norm.

In what follows, we denote by $A_{\Omega,Z}$ the matrix obtained from the observed entries of $A + Z$. This is the input of the recovery algorithm. Beyond the three basic assumptions (see Subsection 1.2), these new works require an extra spectral assumption that either the condition number is small [24] or the gaps between consecutive singular values are large [25].

These assumptions are strong, and it is unclear how often they hold in practice. Figure 1.3 below is based on the Yale face database, a well-known and frequently used dataset; see Wainwright's book [26]. The data matrix in $\mathbb{R}^{165 \times 77760}$ is created by flattening 165 greyscale $243 \times 320$ facial images into row vectors in $\mathbb{R}^{77760}$ and centering them by subtracting the average row. The singular values decay quickly, resulting in a rather large condition number; $\kappa \approx 10.45$ for the first 30 singular values. Consequently, the gaps between singular values are generally large, but there are still exceptions (for example, at index 15).

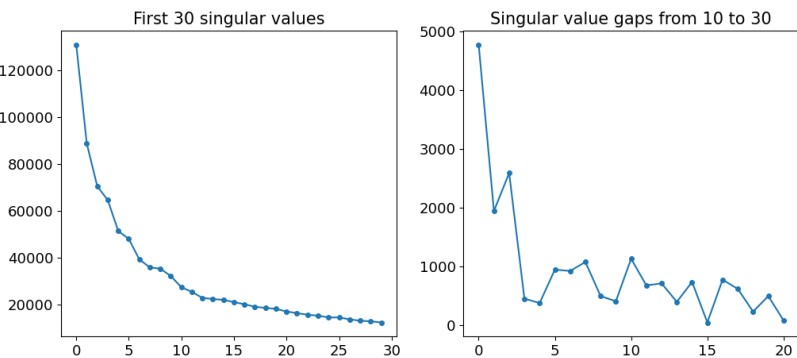

Figure 1: The Yale face database spectrum and spectral gaps

From a mathematical viewpoint, we observe a curious phenomenon: these two extra assumptions are seemingly contradictory. The first (small condition number) indicates that the singular values should be close to each other, while the second requires them to be apart. It is thus natural to conjecture that neither of these conditions is necessary.

In this paper, we prove this conjecture. We show that a properly chosen low-rank approximation of $A_{\Omega,Z}$ approximates $A$ (with the required precision) in the infinity norm, using only the three basic assumptions. This not only provides a mathematically satisfying answer but also significantly expands the range of applications (regarding real datasets).

Our analysis of the algorithm is based on novel mathematical developments and is entirely distinct from previous approaches. It combines the contour integral method introduced in Tran and Vu [27] with a novel bound on random matrix powers, in a fairly nontrivial manner. This approach is robust, and we believe in its potential for future applications.

## 2 New results: a unifying exact recovery method

### 2.1 Formal setting and algorithm

To state the algorithm and our main result, let us restate the setting for clarity.

**Setting 2.1** (Matrix completion with noise). Consider the gound truth matrix $A$, the observed set $\Omega$, and noise matrix $Z$. We assume: (1) $\|A\|_\infty \leq K_A$ for a known parameter $K_A$; (2) we know an upper bound $r_{\max} \geq r$, without needing to know $r$; and (3) the noise $Z$ has independent entries satisfying $\mathbf{E}\left[Z_{ij}\right] = 0$ and $\mathbf{E}\left[|Z_{ij}|^l\right] \leq K_Z^l$ for all $l \in \mathbb{N}$ for a known parameter $K_Z$, without necessarily having the same distribution or variance. The parameters $r$, $r_{\max}$, $K_A$, $K_Z$ can depend on $m$ and $n$.

**Algorithm 2.2** (Approximate-and-Round 2). Input: the $m \times n$ sample matrix $A_{\Omega,Z}$ and the discretization unit $\varepsilon$ of $A$'s entries.

1. *Empirical rescaling:* Let $\hat{p} := (mn)^{-1}|\Omega|$ and $\hat{A} := \hat{p}^{-1}A_{\Omega,Z}$.

2. *Low-rank approximation:* Compute the truncated SVD $\hat{A}_{r_{\max}} = \sum_{i \leq r_{\max}} \hat{\sigma}_i \hat{u}_i \hat{v}_i^T$.

   Take the largest index $s \leq r_{\max} - 1$ such that $\hat{\sigma}_s - \hat{\sigma}_{s+1} \geq 20(K_A + K_Z)\sqrt{\frac{r_{\max}(m+n)}{\hat{p}}}$.

   If no such $s$ exists, take $s = r_{\max}$. Let $\hat{A}_s := \sum_{i \leq s} \hat{\sigma}_i \hat{u}_i \hat{v}_i^T$,

3. *Rounding off:* Round each entry of $\hat{A}_s$ to the nearest multiple of $\varepsilon$. Return $\hat{A}_s$.

Compared to the algorithm Approximate-and-Round used in [25], a minor difference is that we use an estimate $\hat{p}$ of $p$, which is highly accurate with high probability. Another innovation is a different threshold for the truncated SVD step that does not require knowledge of the parameter $\mu_0$. From a complexity viewpoint, our algorithm is efficient, consisting of a truncated SVD on a matrix with $|\Omega|$ non-zero entries, and a rounding step, requiring only $O(|\Omega|r + mn) = O((pr + 1)mn)$ FLOPs.

Our main theorem below provides sufficient conditions for exact recovery with this algorithm.

**Theorem 2.3.** *There is a universal constant $C > 0$ such that the following holds. Suppose $r_{\max} \leq \log^2 N$. Under the model 2.1, assume that the signal is sufficiently large,*

$$\|A\| = \sigma_1 \geq 100rK\sqrt{\frac{r_{\max}N}{p}},$$

*for $K := K_A + K_Z$; and the sampling is sufficiently dense,*

$$p \geq C\left(\frac{1}{m} + \frac{1}{n}\right)\max\left\{\log^4 N, \frac{r^3 K^2}{\varepsilon^2}\left(1 + \frac{\mu_0^2}{\log^2 N}\right)\right\}\log^6 N. \tag{3}$$

*Then with probability $1 - O(N^{-1})$, the low-rank approximation step of Approximate-and-Round 2 recovers every entry of $A$ within an absolute error $\varepsilon/3$. Consequently, if all entries are multiples integer of $\varepsilon$, the rounding-off step recovers $A$ exactly.*

As promised, our result unifies [24] and [25], removing both the condition number and the gap assumptions. Furthermore, our result also implies a RMSE recovery with the same error margin with no constraint on the condition number, an improvement over [17, 18, 15].

## 2.2 Analysis of the result

We begin with a small experiment to showcase the performance of the algorith. Each of the 13 datapoints corresponds to a pair $(m, n)$ that is either $(2^l, 2^{l-1})$ or $(2^l, 2^l)$ for $l = 10, 11, \ldots, 16$.

The matrix $A \in \mathbb{R}^{m \times n}$ is randomly generated by $A_0 = XY^T$, where $X \in \mathbb{R}^{m \times r}$ and $Y \in \mathbb{R}^{n \times r}$, with iid $N(0, 1)$ entries, then $A := \frac{3}{4} A_0 / \|A_0\|_\infty$. The noise matrix $Z$ is generated with $Z_0 \in \mathbb{R}^{m \times n}$ having iid $N(0, 1/4)$ entries, then set $Z_{ij} = \text{sgn}((Z_0)_{ij}) \max\{1/4, |(Z_0)_{ij}|\}$. This normalization makes $K_A + K_Z = 3/4 + 1/4 = 1$. For the sampling, we fix $p = 0.1$ in all datapoints.

In Figure 2, the plots show $N = m + n$ on the horizontal axis, on a log scale, and the RMSE and infinity norm errors on the vertical axes. Both decline rapidly with $N$. In the last datapoint, where $m = n = 2^{16} \approx 6000$, one can recover every entry of $A$ to within an absolute error of $0.1$.

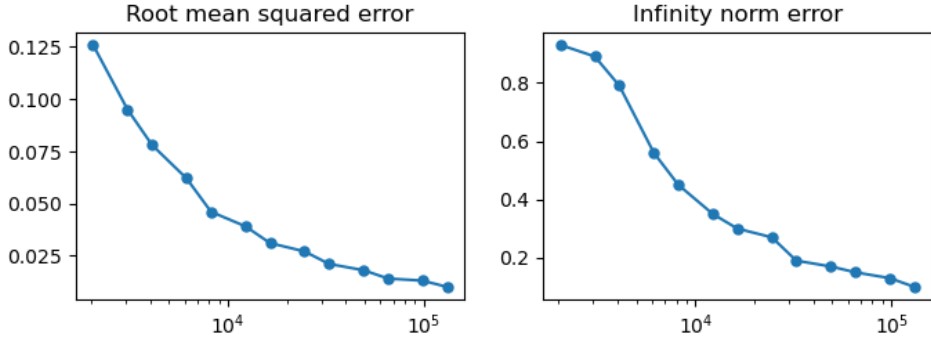

Figure 2: The RMSE and Infinity norm error of our method

Next, let us discuss the optimality of our result from a theoretical view.

**Remark 2.4** (A high-level explanation for the thresholding rule). Note that the observed matrix, rescaled by $p^{-1}$, is a perturbed version of the original matrix, since

$$\mathbf{E}\left[p^{-1} A_{\Omega, Z}\right] = p^{-1}(\mathbf{E}\left[A_\Omega\right] + \mathbf{E}\left[Z\right]) = A.$$

Therefore, one can write $p^{-1} A_{\Omega, Z} = \tilde{A} = A + E$, where $E$ is a random matrix of mean 0 and independent entries. One can view $E$ as a type of "noise" that includes both $Z$ and the noise caused by the random sampling. Since it is a well-known fact in random matrix theory that $\|Z\| = O(K_Z \sqrt{N/p})$ (see [28, 29] for proofs), we simply want to cut off the singular values of $\tilde{A}$ at a level above $\|E\|$, as the well-documented BBP phenomenon [30–35] shows that the part of $A$ below that is fully absorbed by $E$ and becomes indistinguishable from noise. Under the assumptions we make, the signal extracted from the spectrum of $\tilde{A}$ above this threshold is a good approximation of $A$. We use the same argument in Remark 2.5 to show the necessity of the lower bound on $\sigma_1$.

In our theorem, we use only the three basic assumptions of Section 1.2, plus that $\|A\| = \sigma_1$ is large enough. This last assumption seems new, but it is not. It is often hidden in previous papers, and is guaranteed by the fact that $A$ has low rank. Indeed

$$\sigma_1^2 = \|A\|^2 \geq \frac{1}{r} \|A\|_F^2.$$

Consider the representative case when $\|A\|_F^2 = \Theta(mn)$. Then $\sigma_1 \geq r^{-1/2} \|A\|_F \geq c\sqrt{\frac{mn}{r}}$. By the density condition (3) and the condition $r_{\max} \leq \log^2 N$, we have

$$p \geq \frac{r^3 r_{\max} K^2 N \log^4 N}{mn} \implies \sigma_1 \geq c\sqrt{\frac{mn}{r}} \gg (\log^2 N) r K \sqrt{\frac{r_{\max} N}{p}},$$

so the assumption on $\sigma_1$ in our theorem is automatically satisfied.

**Remark 2.5** (Necessity of the signal assumption). From the engineering perspective, it is intuitive that $\sigma_1$ should be large enough in order for any kind of recovery to be possible. In the opposite case when the intensity of the noise $Z$ dominates the signal $A$, then the data is "too corrupted" and all interersting information are lost. Rigorously, the well-documented BBP phenomenon in random matrices [30–35] states that, if $\|Z\| \geq c\|A\| = c\sigma_1$, for a specific constant $c$, then $A + Z$ is indistinguishable from a fully random matrix, leaving no chance to recover $A$ even if $A + Z$ is fully observed. Since it is also a well-known fact in random matrix theory that $\|Z\| = O(K_Z \sqrt{N/p})$ (see [28, 29] for proofs), the condition in Theorem 2.3 is simply $\sigma_1 \geq Cr\|Z\|$, which is optimal when $r = O(1)$.

**Remark 2.6** (Optimality of the density bound). The condition (3) looks complicated, but in the base case where $r = O(1)$, $K_A + K_Z = O(1)$, and uniformly random singular vectors, so that we have $\mu_0 = O(\log N)$, it reduces to

$$p \geq C \max \left\{ \log^4 N, \ \varepsilon^{-2} \right\} \left( m^{-1} + n^{-1} \right) \log^6 N, \tag{4}$$

which is equivalent to $|\Omega| \geq CN \log^6 N \max\{\log^4 N, \varepsilon^{-2}\}$ in the uniform sampling model. The power of $N$ is optimal to the theoretical limit (see Section 1.2). The power of $\log N$ can be further reduced but the details are tedious, and the improvement is not really important from the pratical view point. For recovery with precision $\varepsilon$, this bound grows with $\varepsilon^{-2}$, which is comparable to or better than all previous works (see [19, Section A]).

**Remark 2.7** (Relaxing the bound on $r_{\max}$). The condition $r_{\max} \leq \log^2 N$ in Theorem 2.3 can be avoided, at the cost having a more complicated sampling density bound, which connects $p$ and $r$

$$p \geq C \left( \frac{1}{m} + \frac{1}{n} \right) \max \left\{ \log^{10} N, \ \frac{r^4 r_{\max} \mu_0^2 K^2}{\varepsilon^2}, \ \frac{r^3 K^2}{\varepsilon^2} \left( 1 + \frac{\mu_0^2}{\log^2 N} \right) \left( 1 + \frac{r^3 \log N}{N} \right) \log^6 N \right\}. \tag{5}$$

The proofs of Theorem 2.3 and Eq. (5) will be in the supplementary appendix, [19, Section B] . This shows that our result does not require any extra condition besides the mandatory large signal assumption.

In Section 3, we give a proof sketch for Theorem 2.3, asserting the correctness of our algorithm, Approximate-and-Round 2. The proof will boil down to obtaining a sharp bound in the infinity norm for the perturbation of the low-rank approximations, for which we introduce Theorem 7. By reframing the problem from a matrix perturbation perspective, we can view Theorem 7 as an infinity norm version of the classical Davis-Kahan-Wedin theorem.

We next explain the main ideas behind its proof, which combine the contour integral technique of Tran and Vu [27] (with necessary adjustments) and a novel *semi-isotropic* bound on powers of a random matrix. The (rather complex) detailed proofs will appear in the supplementary appendix, [19, Section C] .

# 3 Main ideas of the proof

## 3.1 The matrix perturbation perspective and the rise of the extra assumptions

Consider $\tilde{A}_s - A_s$. Let $\rho := \hat{p}/p$ and $\tilde{A} = p^{-1} A_{\Omega,Z} = \rho^{-1} \hat{A}$, we can write

$$\hat{A}_s - A = \rho^{-1} \tilde{A}_s - A = (\rho^{-1} - 1)A + \rho^{-1}(A_s - A) + \rho^{-1}(\tilde{A}_s - A_s).$$

We show that the three error terms on the right-hand side are small in the infinity norm. The first is easy, since $\rho$ is close to 1 (by a Chernoff bound, see e.g. Hoeffding [36]). For the second, we have

$$\|A - A_s\|_\infty = \left\| \sum_{i \geq s+1} \sigma_i u_i v_i^T \right\|_\infty \leq \sigma_{s+1} \|U\|_{2,\infty} \|V\|_{2,\infty} \leq \frac{r \sigma_{s+1} \mu_0}{\sqrt{mn}},$$

which will be small by the way we choose $s$ and the incoherence property. Most of the heavy lifting goes to bounding the third term, $\|\tilde{A}_s - A_s\|_\infty$.

Observe that $\mathbf{E}[A_\Omega] = pA$ and $\mathbf{E}[Z_\Omega] = 0$, so that $\mathbf{E}[\tilde{A}] = p^{-1}\mathbf{E}[A_\Omega + Z_\Omega] = A$. Therefore, $E := \tilde{A} - A$ is a random matrix with mean 0. This opens a way to use tools and ideas from random matrix theory in the analysis.

The approaches in [24, 25] both arrive at a bound on $\|\tilde{A}_s - A_s\|_\infty$, but at the cost of their respective extra assumptions. We give a brief overview of their methods.

Suppose $s = r = 1$ for simplicity, one can write

$$\tilde{A}_1 - A_1 = \tilde{\sigma}_1 \tilde{u}_1 \tilde{v}_1^T - \sigma_1 u_1 v_1^T = (\tilde{\sigma}_1 - \sigma_1) u_1 v_1^T + \sigma_1 \left( \tilde{u}_1^T \tilde{v}_1^T - u_1 v_1^T \right)$$
$$= (\Delta \sigma_1) u_1 v_1^T + \sigma_1 (\Delta u_1) v_1^T + \sigma_1 u_1 (\Delta v_1)^T + \sigma_1 (\Delta u_1)(\Delta v_1)^T,$$

where we set $\Delta \sigma_1 := \tilde{\sigma}_1 - \sigma_1$ and analogously for other $\Delta$-notations.

By Weyl's inequality, $|\Delta \sigma_1| \leq \|E\|$, so the first term is bounded by $\|u_1\|_\infty \|v_1\|_\infty \|E\|$ in the infinity norm. The main challenge is to bound the middle two terms, which dominate the last one. By symmetry, let us focus on $\sigma_1 (\Delta u_1) v_1^T$. We have

$$\|\sigma_1 (\Delta u_1) v_1^T\|_\infty = \sigma_1 \|\Delta u_1\|_\infty \|v_1\|_\infty.$$

If $\sigma_1$ is large enough compared to $\|E\|$, [25] showed that the error $\Delta u_1$ is sufficiently "spread out", namely, $\|\Delta u_1\|_\infty \leq C \|\Delta u_1\| \|u_1\|_\infty$. To bound $\|\Delta u_1\|$, the classic **Davis-Kahan-Wedin theorem** [37, 38] gives $\|\Delta u_1\| \leq C\|E\|/\sigma_1$, so the final bound on this term looks like

$$\|\sigma_1 (\Delta u_1) v_1^T\|_\infty \leq C \|u_1\|_\infty \|v_1\|_\infty \|E\|.$$

Putting everything together, one obtains the desired (optimal) bound

$$\|\tilde{A}_1 - A_1\|_\infty \leq C \|u_1\|_\infty \|v_1\|_\infty \|E\| \leq \frac{C \mu_0}{\sqrt{mn}} \|E\|. \tag{6}$$

When extending this argument to rank $s > 1$, [24] and [25] used two different approaches. In [24], the authors wrote

$$\tilde{A}_s - A_s = U_s (\Delta \Sigma_s) V_s^T + (\Delta U_s) \Sigma_s V_s^T + U_s \Sigma_s (\Delta V_s)^T + (\Delta U_s) \Sigma_s (\Delta V_s)^T.$$

Again, one needs to bound $\|(\Delta U_s) \Sigma_s V_s^T\|_\infty$. The problem here is that $\|\Sigma_s\|$ is still $\sigma_1$ (instead of $\sigma_s$), which eventually leads to the appearnce of the condition number $\kappa = \sigma_1/\sigma_r$ in the final bound. Furthermore, the algorithm in [24] requires knowledge of the rank $r$ of $A$. This is often not the case in practice. However, Keshavan et al. [17] showed that one can estimate $r$ precisely with high probability, given that $\kappa = O(1)$. Thus, the assumption that the condition number is small is needed here for two different reasons: to compute the rank and to improve the quality of the bound; see also [19, Section A].

[25] circumvented this issue by considering

$$\tilde{A}_s - A_s = \sum_{i \leq s} (\tilde{\sigma}_i \tilde{u}_i \tilde{v}_i^T - \sigma_i u_i v_i^T) = \sum_{i \leq s} \Delta (\sigma_i u_i v_i^T),$$

then bounding each term $\Delta(\sigma_i u_i v_i^T)$ separately. Again, this boils down to bounding $\|\sigma_i (\Delta u_i) v_i^T\|_\infty$. For this, they use a (stronger) variant of Davis-Kahan theorem proved in O'Rourke et al. [39] (see [19, Section B]). As a trade-off, this approach requries the assumption that the singular values $\sigma_i$ are well-separated for the (stronger) Davis-Kahan bound to hold.

Our new method, which is entirely different, allows us to avoid both assumptions. Let us first establish the quantitative statement:

**Theorem 3.1.** *Consider a deterministic matrix $A \in \mathbb{R}^{m \times n}$. and a random matrix $E \in \mathbb{R}^{m \times n}$ with independent entries satisfying $\mathbf{E}[E_{ij}] = 0$ and $\mathbf{E}\left[|E_{ij}|^l\right] \leq p^{1-l} K^l$ for some $K > 0$ and $0 < p < 1$. Let $\tilde{A} = A + E$. Let $s \in [r]$ be an index satisfying*

$$\delta_s := \sigma_s - \sigma_{s+1} \geq 40 r K \sqrt{N/p},$$

*There are constant $C, C_1$ such that, if $p \geq C(m^{-1} + n^{-1}) \log N$ (where $N = \max\{m, n\}$), then*

$$\|\tilde{A}_s - A_s\|_\infty \leq \frac{C_1 (\mu_0 + \log N) \log^2 N}{\sqrt{mn}} r \sigma_s \left( \frac{K \sqrt{N}}{\sigma_s \sqrt{p}} + \frac{r K \sqrt{\log N}}{\delta_s \sqrt{p}} + \frac{r^2 \mu_0 K \log N}{p \delta_s \sqrt{mn}} \right). \tag{7}$$

From Eq. (7), one can verify, with a routine computation, that the sampling density condition (3), with a sufficiently large constant $C$, implies $\|\tilde{A}_s - A_s\|_\infty \leq \varepsilon/3$, which proves Theorem 2.3. For a detailed proof of Theorem 2.3, see [19, Section B].

## 3.2 Our new approach: contour integrals and bounds on random matrix powers

Let us now elaborate on our new approach used to prove Theorem 3.1.

We begin with the contour integral argument, adopted from the technique by P. Tran and Vu [27]. Instead of analyzing $\tilde{A}_s - A_s$ directly, we turn to their *symmetrized versions*. Denote

$$B_{\text{sym}} = \begin{bmatrix} 0 & B \\ B^T & 0 \end{bmatrix}$$

for any matrix $B$, we have $\tilde{A}_{\text{sym}} = A_{\text{sym}} + E_{\text{sym}}$. Note that the symmetrization preserves the spectral norm and transforms the singular value decomposition into the eigendecomposition

$$A_{\text{sym}} = W\Lambda W^T = \sum_{|i| \in [r]} \sigma_i w_i w_i^T,$$

where for each $i \in [r]$,

$$w_i = \frac{1}{\sqrt{2}} \begin{bmatrix} u_i \\ v_i \end{bmatrix}, \qquad w_{-i} = \frac{1}{\sqrt{2}} \begin{bmatrix} u_i \\ -v_i \end{bmatrix}, \quad \sigma_{-i} = -\sigma_i.$$

The first key idea here is to compare $\tilde{A}_{\text{sym}}$ and $A_{\text{sym}}$ via their *Stieltjes transforms*. Let $z$ be a complex variable, we have the expansion (we pretend that convergence is not an issue for now)

$$z(zI - \tilde{A}_{\text{sym}})^{-1} - z(zI - A_{\text{sym}})^{-1} = \sum_{\gamma=1}^{\infty} z\left[(zI - A_{\text{sym}})^{-1} E_{\text{sym}}\right]^\gamma (zI - A_{\text{sym}})^{-1}.$$

If we integrate both sides over a contour $\Gamma_s$ which encloses only the eigenvalues $\{\sigma_i : |i| \in [s]\}$, we obtain, by Cauchy's integration theorem (and some light calculation),

$$(\tilde{A}_s - A_s)_{\text{sym}} = \sum_{\gamma=1}^{\infty} \oint_{\Gamma_s} \frac{z\,\mathrm{d}z}{2\pi i} \left[(zI - A_{\text{sym}})^{-1} E_{\text{sym}}\right]^\gamma (zI - A_{\text{sym}})^{-1}.$$

Using the formula $(zI - A_{\text{sym}})^{-1} = \sum_{|i| \in [r]} (z(z-\sigma_i))^{-1} w_i w_i^T + z^{-1} I$, we can expand the right-hand side (via some calculations) to obtain

$$(\tilde{A}_s - A_s)_{\text{sym}} = \sum_{\gamma=1}^{\infty} \sum_{*,*,\ldots,*} \mathcal{C}(*, *, \ldots, *) E_{\text{sym}}^* w_* w_*^T E_{\text{sym}}^* w_* w_*^T \ldots w_* w_*^T E_{\text{sym}}^* w_* w_*^T E_{\text{sym}}^*,$$

where the asterisks stand for integer variables, i.e. $w_*$ can be any $w_i$, $E_{\text{sym}}^*$ can be any positive power of $E_{\text{sym}}$, and $\mathcal{C}$ is a scalar coefficient of the form

$$\mathcal{C}(*, *, \ldots, *) = \oint_{\Gamma_s} \frac{z\,\mathrm{d}z}{2\pi i} \frac{1}{z^*(z - \sigma_*)(z - \sigma_*)\ldots(z - \sigma_*)}.$$

There are conditions on these variables, but we will overlook them for now. At this point, one can bound $|\tilde{A}_s - A_s|$ by bounding the terms above and applying the triangle inequality. These bounds are fairly technical and require some delicate combinatorial and analytical consideration. This is essentially the approach introduced in [40].

Since we aim for the infinity norm, which is much harder to deal with than the spectral norm, we need to significantly modify the above argument.

Let us consider, for example, the 11-(upper left corner) entry of the matrix in question. We have

$$(\tilde{A}_s - A_s)_{11} = \sum_{\gamma=1}^{\infty} \sum_{*,\ldots,*} \mathcal{C}(*, *, \ldots, *)\left(e_{m+1}^T E_{\text{sym}}^* w_*\right)\left(w_*^T E_{\text{sym}}^* w_* w_*^T \ldots w_*^T E_{\text{sym}}^* w_*\right)\left(w_*^T E_{\text{sym}}^* e_1\right),$$

where $e_1, e_2, \ldots$ are standard basis vectors in $\mathbb{R}^N$. The upper left entry of a matrix $M$ is $e_1^T M e_1$, but notice that we use $e_{m+1}$ on the left due to the symmetrization.

To bound $\|\tilde{A}_s - A_s\|_\infty$, we need a strong bound on $e_j^T E_{\text{sym}}^a w_i$, for all indices $j$. To this end, we prove a new *semi-isotropic bound* for powers of $E_{\text{sym}}$, which is a key technical contribution of this paper.

These bounds take advantage of the fact that $E$ has independent entries. The exact bounds are a bit technical, but for the simple case when $\mu_0 = O(1)$, they take the form

$$|e_j^T E_{\text{sym}}^a w_i| \leq \begin{cases} C \log N \sqrt{\frac{\mu_0 + \log N}{m}} \|E\|^a & \text{for } 1 \leq j \leq m, \\ C \log N \sqrt{\frac{\mu_0 + \log N}{n}} \|E\|^a & \text{for } m + 1 \leq j \leq N. \end{cases}$$

By setting $a = 0$, one can also see that these bounds are optimal, up to $\log N$ factors. Plugging these bounds into the expansion, one can eventually obtain the bound of Theorem 3.1, by modifying the steps from [27] in a proper manner. This is, in itself, a challenging task, and we provide the full detailed proof in the supplementary appendix [19, Section C] .

Note that there have been similar isotropic-style bounds for powers of random matrices, notably by Mao et al. [41] and Fan et al. [42]. They both cover a wide range of cases, but were designed optimally for the use cases in their respective papers. In the supplementary document [19, Section B], we will briefly explain that they are not strong enough in our use case.

Besides the bound on the perturbation of low-rank approximations, we obtain similar bounds for the perturbation of singular vectors, in both the infinity and 2-to-$\infty$ norms. For instance, we can show

$$\|\tilde{U}_s \tilde{U}_s^T - U_s U_s^T\|_\infty \leq \frac{C \log^2 N (\mu_0 + \log N)}{m} \left( \frac{\|E\|}{\sigma_s} + \frac{r\|U^T E V\|}{\delta_s} \right)$$

$$\|\tilde{U}_s \tilde{U}_s^T - U_s U_s^T\|_{2,\infty} \leq C \log N \sqrt{\frac{(\mu_0 + \log N)}{m}} \left( \frac{\|E\|}{\sigma_s} + \frac{r\|U^T E V\|}{\delta_s} \right)$$

under the assumptions of Theorem 3.1. We will use these bounds in a future paper.

## 3.3 Summary and roadmap for the rest of the paper

We started with the famous and influential matrix completion problem (Section 1.1, its three basic assumptions. Next, we discussed the the noisy setting, which is more realistic (Section 1.2, and focused on recent results concerning exact recovery [24, 25] (Section 1.3). These results require extra assumptions on the spectrum of the ground matrix. On one hand, these assumptions are rather strong and thus significantly limit the application of these results on real data sets. On the other hand, they, quite intriguingly, seem to contradict each other from the mathematical view point. This leads to the conjecture that neither is needed.

We next introduced our own result, which provides an efficient algorithm without requiring the above mentioned assumptions (Section 2), showing that the conjecture is correct. This algorithm obtains exact recovery under only three basic assumptions, and is the first such algorithm for noisy data. In the (easier) noiseless case, the only algorithm using only three basic assumptions is that of Candes et al. [13, 4, 14], which is based on convex optimization. Compared to this algorithm, our is simpler and faster, as it uses only on round of low rank approximation. (Low rank approximation is known to be an efficient operator, used very often in practice.) Thus, our result makes a contribution in the noiseless case as well.

Our main theorem is Theorem 2.3, which guarantees the correctness of our algorithm. This, in essence, is a matrix perturbation bound, and can be seen as an infinity norm variant of the classical Davis-Kahan theorem.

In Section 3, we sketch the proof of Theorem 2.3. We started with the sketch of the arguments of [24, 25], and showed why their extra assumptions are required. Next, we describe our new approach (Section 3.2), which combines the contour integration method intorduced in [27] with novel semi-isotropic bounds for random matrix powers. This approach avoids the use of the extra assumptions in earlier papers. Over all, this has led to a highly non-trivial, but robust and powerful, machinery to obtain matrix perburtation bounds in the infinity norm. We believe that this method will have many other applications.

The supplementary appendix [19], separate from the main body, will contain four sections.

In [19, Section A] , we dive deeper into the technical bounds of other matrix completion papers, including [24, 25] and the RMSE recovery papers, pointing out the strong assumptions they use (which mostly requiring the condition number of the ground matrix to be small), and demonstrate

that we do not need those assumptions. Another point of interest is the sample size needed to recover $A$ with precision $\varepsilon$ (or within a RMSE $\varepsilon$), which should grow with $1/\varepsilon$. We demonstrate that our growth factor, which is $1/\varepsilon^2$, is on par with the best of these results.

Consider the simple setting where $K_Z = O(1)$, $\mu_0 = O(\log N)$ and $r = O(\log^2 N)$. Table 1 summarizes the advantages we have over the main methods discussed in this paper.

| | Method | Entry-wise recovery? | Time Complexity | Extra assumption to achieve optimal sampling bound |
|---|---|---|---|---|
| Convex Optimization | Candes and Plan | No | $O(\lvert\Omega\rvert^2(m+n)^2)$ | Not optimal |
| Low-rank approx. with grad. descent cleaning up | Keshavan et al. | No | $O(\lvert\Omega\rvert r + mn + Lr(m+n))$, if one chooses $L$ iterations for the clearning step | Condition number is small |
| Single-step low-rank approx. with singular value thresholding | Abbe and Fan | Yes | $O(\lvert\Omega\rvert r + mn)$ | Condition number is small |
| | Bhardwaj et al. | Yes | $O(\lvert\Omega\rvert r + mn)$ | Every singular value gap is large |
| | Our method | Yes | $O(\lvert\Omega\rvert r + mn)$ | None |

Table 1: Comparison of methods for noisy matrix completion

In [19, Section B], we introduce our main technical machinery behind the proof of Theorem 3.1. We will fully shift the context to matrix perturbation, and introduce new notation to be used throughout the ensuing discussion and proofs. We will have one main theorem for the contour integral method [19, Theorem B.2], another for the semi-isotropic bounds [19, Theorem B.4], and a corollary of their combination [19, Theorem B.6]. We then use this theorem to prove Theorem 3.1, then prove Theorem 2.3 with Theorem 3.1.

In [19, Section C], we provide the full proofs of the main technical theorems. To assist the readers, these proofs come with their own sketches. In particular, the proof sketch of [19, Theorem B.2] will be a more detailed version of the sketch in Section 3.2. All details will be presented, barring some cumbersome technical lemmas.

Finally, in [19, Section D], we prove the technical lemmas. The details are heavy, but we hope that the revealed intution earlier will significantly help the readers follow the proofs.

## Acknowledgments and Disclosure of Funding

The research is partially supported by Simon Foundation award SFI-MPS-SFM-00006506 and NSF grant AWD 0010308.

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
