# OpenReview forum: "Fast exact recovery of noisy matrix from few entries: the infinity norm approach"
_NeurIPS.cc/2025/Conference — NeurIPS 2025 poster_

### Official Review · Reviewer_nrNZ · 2025-06-11

**Clarity:** 3
**Significance:** 1
**Originality:** 2
**Rating:** 2
**Confidence:** 4

**Summary:**

This manuscript considers the problem of exact recovery of a low rank matrix from noisy observations, under an additional assumption that
the underlying matrix entries have fixed precision (i.e. they are all integer multiples of some epsilon).

The authors study a simple SVD-based algorithm followed by rounding. They prove that, under certain conditions, their method can exactly recover the target matrix even in the presence of additive noise. The analysis combines tools from random matrix theory and from matrix perturbation.

**Questions:**

- P4 L127 the authors write "this not only provides a mathematically ... .but also significantly expands the range of applications (regarding real datasets). Can the authors provide convincing examples for this claim ?

- Can the authors discuss how to estimate the parameters K_A and K_Z in the proposed algorithm ?

- P5 L160 RMSE recovery with the same error margin. Can you please explain this sentence ?

**Ethical Concerns:**

["NO or VERY MINOR ethics concerns only"]

**Final Justification:**

In my opinion this manuscript has potential to make a significant impact. However, in its current form, it considers a very niche and not particularly interesting problem (exact recovery of low rank matrices with finite precision). It does not cite nor discuss the possible relation and implications of this work to the more recent literature on exact recovery guarantees for various methods. Overall, my assessment remains the same. In my opinion this manuscript requires significant and major revisions and a careful re-evaluation before being suitable for publication at a top venue such as NeurIPS.

**Limitations:**

This work is theoretical and has no societal impact.

**Quality:**

2

**Strengths And Weaknesses:**

Strengths:
The use of sophisticated tools from random matrix theory enables the authors to establish recovery guarantees that surpass existing
results in similar settings.
The paper is relatively clearly written and accessible.

Weaknesses:
The problem considered in this manuscript is uncommon and non-standard in the matrix completion literature.
In terms of originality, the method considered is (up to the rounding step and how to choose the number of components) standard as an initialization in the matrix completion literature, and so there is limited novelty here.
These issues raise major concerns regarding the importance of the problems studied, the significance of their findings for the community and their practical relevance.

While the authors motivate their work by practical applications, the manuscript lacks an empirical evaluation to support the theoretical claims or illustrate the method's performance in practice.

Fit to NeurIPS: as the authors write, one of their theorems can be seen as an infinity norm variant of the classical Davis-Kahan Theorem. Perhaps a more suitable venue for this work would be a journal focused on matrix analysis.

Additional comments:
- Problem formulation: the authors assume the underlying matrix has finite precision while observations contain noise. It would strengthen the manuscript if the authors were to provide concrete examples where this setting holds. To the best of my knowledge, the more common setting is one in which the observations are quantized (e.g, have finite precision) with the underlying matrix assumed low rank but not with bounded precision. Examples include movie ratings (e.g. observed values are 1-5) with various stochastic models relating how observed ratings are related to the underlying matrix entries, and the 1-bit matrix completion problem, where observed entries are +1/-1, again where they underlying matrix is assumed low rank.

- Abstract is very long. In addition, the abstract cites various papers, which is rather uncommon.

- The authors write that the paper by Abbe et al [9] showed that one can recover A even in the noisy case, provided A has bounded precision [p1 L15], and a similar statement on p3 L101. This seems to imply that [9] also considered the precise problem studied in this manuscript of exact recovery in a matrix completion problem. Unless I missed something, I find this rather misleading. Ref. [9] discussed exact recovery for stochastic block models, which is a different problem. To the best of my understanding, Section 3.3 of [9], titled matrix completion with noisy entries does not assume bounded precision.

- p1 L32 to be precise, the task is to recover A given A_\Omega *as well as* the set Omega itself.

- p1 L66 the authors claim that many papers assume r is bounded while m,n\to\infty. I find this inaccurate for the matrix completion literature. It often holds in mathematical statistics / RMT papers.

- p3 L113-end: the authors show the eigenvalue decay for the Yale face dataset. The relation between this example and the manuscript is unclear. At best, this result shows that many practical datasets do not have an exact low rank, but at best are only approximately low rank.

- A small remark is that matrix completion becomes challenging when the number of observed entries is very small and yet above the information limit. In this case, the SVD of the observed entries is not sufficiently accurate, and yet there are various methods that empirically are still able to recover the matrix.

---

> ### Author Rebuttal · Authors · 2025-07-30
>
> Thank you for the through and detailed comments and the evaluation of our paper. We would like to offer our responses to some of your concerns and questions.
>
> **1. Comments**
>
> 1.  Regarding the lack of experiments on real datasets: we thank you for pointing it out, and we agree that we should have included a couple. Other reviews also pointed out that our assumption of low rank still does not fully reflect real-life datasets, which are often only approximate low-rank. On the other hand, most breakthrough mathematical works like Candes-Tao's make use of the low rank assumption.
> Initially, we wanted to focus on the fact our result breaks the two theoretical barriers involving the condition number and large singular value gaps as the main highlight. We will certainly include experiments, paralleling those in other works in the final version. We are also working on an extension of this paper to cover the case when the ground matrix is approximately low rank.
>
> 2. Regarding the comparison with Abbe et al. (reference [9] in our paper): We felt that, at least in Section 3.3 of [9], the problems both of us were trying to solve are essentially the same, as we both aimed to provide an infinity norm estimate for the error of our respective recovery methods in terms of $n$, $p$ and other parameters from the underlying matrix $A$, which can then be turned to an estimate on the minimum sampling rate ($p$) needed to make this error within a chosen $\varepsilon$. We only need to look at the precision level of the entries of $A$ if we want to round off the output to achieve exact recovery (one can imagine that the output before rounding off has higher precision level, for example, 64-bit floating point numbers while $A$'s entries are half-integers).
> As the focus of [9] is a different problem,
> we appologize for making the comparison with respect to matrix completion. We can omit this comparison and will only have  a   comparison with Section 3.3, and with respect to the infinity norm bounds.
>
> 3. Regarding the inaccuracies on page 1: We thank you for pointing them out, and we will certainly fix them in the final version. We would also like to note that our result holds for a general $r$ that can go to infinity with $m$ and $n$.
>
> 4. Regarding the comment about page 3: We refer to our first paragraph, which admits the limit of our current scope to low-rank matrices.
>
> **2. Questions**
>
> 1. Regarding page 4 line 127: We meant to explain that our result guarantees (with high probability) that the method of truncated SVD can now be applied to recover, down to every entry, matrices with large condition numbers, and matrices with some small singular value gaps, which were not guaranteed by previous methods. As mentioned above, we will include experiments with some datasets in the final version.
>
> 2. Regarding how to estimate $K_A$ and $K_Z$: We simply assume they will be given along with the dataset. For example, $K_A + K_Z \le 5$ in the Netflix problem. Datasets often come with descriptions about the range of values each cell can take.
>
> 3. Regarding page 5 line 160: We meant to explain that if we obtain a recovery within $\varepsilon$ in the infinity norm, meaning $||\hat{A} - A||\_\infty \le \varepsilon$, then this recovery is also within $\varepsilon$ in the root-mean-squared-error sense, namely $\frac{1}{\sqrt{mn}} ||\hat{A} - A||_F \le \varepsilon$, which is a fairly straightforward observation. We apologize for the confusion caused by our word choices, and will add a couple sentences to clarify this in the final version.

---

> > ### Comment · Reviewer_nrNZ · 2025-08-05
> >
> > I thanks the referees for their reply.
> > A fundamental question is whether the results of this work could improve existing exact recovery guarantees for other methods for matrix completion that do not assume the matrix to have finite precision. For example, various factorization based methods have guarantees for exact recovery provided the starting point is sufficiently accurate.
> > It may be interesting to consider this issue / discuss it in the manuscript.

---

> > > ### Author Response · Authors · 2025-08-07
> > >
> > > We thank you for raising the question of whether our results could improve the exact recovery results without the finite precision assumption. It is indeed a very interesting and worthy question. As of right now, the methods in this regime that we are aware of are all iterative, whose output converges to the ground truth matrix $A$ as the number of iterations goes to infinity. We believe our method can at least help some of these methods choose a more accurate starting point, or help proving that their starting points are more accurate than previously thought, and thus potentially improve their guarantees for recovery. We will discuss this potential application in our final version.
> > >
> > > It is also worth noting that the lines of work in the non-finite precision regime that we are aware of, namely by Hardt et al. and Keshavan et al., do not show that exact recovery is possible in finite time. The former gave a relative error bound in the Frobenius norm and the number of steps to reach any error margin, while the latter did not discuss time complexity. An approximate close in the Frobenius norm does not necessarily close in the infinity norm, so there might be entries of $A$ that are not recovered. On the other hand, in the bounded precision case, our results guarantee an upper bound on the time to recover $A$ exactly entry-wise. Without any assumption on the precision, we still have an arbitrarily close approximation in  the infinity norm. However, it is likely that we have missed several recent works that achieve exact recovery in time that are finite and shorter than that of nuclear norm minimization. We will certainly conduct a more thorough literature review and include more discussion in the final version.

---

> > > > ### Comment · Reviewer_nrNZ · 2025-08-07
> > > >
> > > > I thank again the referees for their reply. I don't understand the sentence in the reply "do not show that exact recovery is possible in finite time". In practice, optimization algorithms perform computations in finite precision. The common approach to study the time complexity of iterative algorithms is to prove they have say linear or quadratic convergence. Hence, they may reach arbitrary precision epsilon within very few iterations (O(log(1/epslion) ) or even less). If such a guarantee holds, then for all practical purposes such a  method reaches exact (up to say machine precision) recovery in "finite time".
> > > >
> > > > The papers by Hardt and Keshavan are over 10 years old. There are several iterative algorithms for matrix completion for which such exact recovery guarantees with linear or quadratic convergence have been derived. It seems from the author's response, and the papers cited in the submitted manuscript. that they are not aware of this newer and probably relevant literature. See for example
> > > > Zheng and Lafferty, 2016,
> > > > Sun and Luo 2018,
> > > > several papers by Yuxin Chen and co-authors from around 2020 and onwards,
> > > > as well as more recent works, which you may easily find by searching for more recent papers that cite these works.

---

> > > > > ### Author Response · Authors · 2025-08-08
> > > > >
> > > > > We would like to thank you again for further clarifications on your comments and pointing out where our previous replies sounded confusing. From the phrase "do not assume the matrix to have finite precision" in your first reply, we were under the impression that you were discussing a setting where it is impossible to round up the output to obtain $A$, and thus no approximation can recover it, which led to our statement that the cited works did not guarantee exact recovery in finite time. We now realized that the setting you mentioned should be that entries of $A$ have a precision, and thus can theoretically be recoverable by a close enough approximation, but that may be too small for certain algorithms, including ours to practical. Theoretically, iterative approaches with fast convergence may be more ideal than single-step SVD because they can improve their approximations by having more iterations instead of asking for more observations like ours do (as our main theorem requires that the sample density grows with $\varepsilon^{-2}$ get within $\varepsilon$ infinity norm error). We thank you for making us aware of the works by Zheng and Laffery, Sun and Luo, Chen et al., and motivating us to follow more recent development. We will examine their results and compare with ours in the final version.
> > > > >
> > > > > As for the works by Hardt et al. and Keshava et al., they both require more samples and more iterations to achieve better approximations. The sample density requirement in Keshavan et al. also grows with $\varepsilon^{-2}$ like ours, but it is quite interesting that Hardt et al. only needed it to grow with $\log^2(1/\varepsilon)$. Compared with the latter, our advantages mainly lie in the simplicity of our algorithm and the fact that our approximation is close uniformly over all entries, while Hardt et al. obtains a relative error in the infinity norm. This means we can easily argue that a simple rounding step recovers $A$ exactly (down to every entry) in our case, but it is less clear how to achieve this in the latter (how many iterations to run and how many samples are needed). As we have mentioned in our first response, we will include some experiments with real data and compare the results with the other methods in the final version to ensure that our method can recover $A$ quickly in practice too.

---

### Official Review · Reviewer_AT2W · 2025-06-27

**Clarity:** 3
**Significance:** 3
**Originality:** 3
**Rating:** 5
**Confidence:** 3

**Summary:**

The paper analyzes the performance of the proposed approximate-and-round 2 algorithm for matrix completion. By leveraging contour integrals and refined bounds on bilinear forms of matrix powers, it eliminates the dependence on the condition number and singular value gaps in the recovery guarantees and establishes an infinity-norm bound between the estimated matrix and the ground truth.

**Questions:**

1. Lines 261-272 suggest that a fine-grained bound on $\left|e_j^{\top} E_{\text {sym}}^a w_i\right|$ is needed. Such bounds are common in the literature, for example, in Section C.4 of Estimating Mixed Memberships with Sharp Eigenvector Deviations by Mao et al. (JASA 2021) and Lemmas 4 and 5 of Asymptotic Theory of Eigenvectors for Random Matrices With Diverging Spikes by Fan et al. (JASA 2022). (I am referring to the arXiv versions of these papers.)
    If I understand correctly, the bounds in Lemmas D.3 and D.4 in the Appendix could be derived using bounds on $\left|e_j^{\top} E_{\text {sym}}^a w_i\right|$. Could you briefly discuss whether the results in these papers can be applied directly to obtain Lemmas D.3 and D.4, or clarify if additional arguments are needed?
2. In the proof of Theorem 2.3, the signal-to-noise assumption changes from $\sigma_1 \geq 100 r K \sqrt{r_{\max } N / p}$ to $\sigma_1 \geq 100 r \kappa \sqrt{r_{\max } N}$. It appears that you are implicitly using $\kappa=K / \sqrt{p}$, which conflicts with the notation where $\kappa$ denotes the condition number of the matrix $A$. Perhaps this is a typo, and you intended to use a different symbol, such as $\varsigma$ as introduced in Theorem B.4?

**Ethical Concerns:**

["NO or VERY MINOR ethics concerns only"]

**Final Justification:**

At the outset, I would like to highlight that the dependence on the condition number in low-rank matrix estimation has long been regarded as an artifact of the proof technique; eliminating this dependence constitutes an important theoretical contribution of the paper. The authors have addressed most of my questions. As a minor point, while the newly introduced matrix concentration inequalities are interesting, I encourage the authors to make clearer connections between these results and the existing literature. Overall, I maintain my positive score of the paper.

**Limitations:**

yes

**Paper Formatting Concerns:**

No formatting concern.

**Quality:**

3

**Strengths And Weaknesses:**

**Strength:**

As mentioned in the summary, the paper presents a novel result on the performance of the approximate-and-round 2 algorithm for matrix completion, removing the dependence on the condition number and singular value gaps in the recovery guarantees. This is a meaningful contribution that offers new theoretical insights to the community. While I was not able to check all the details of the proofs due to time constraints, the overall approach appears solid and well-motivated.

**Weaknesses:**
1. I recommend adding numerical experiments to demonstrate that the algorithm's performance does not depend on the condition number $\kappa$ if time permits.
2. The assumption that $A$ has a distortion level of $\varepsilon$ feels somewhat unnatural. It effectively implies that $A=$ $\varepsilon B$ for some integer matrix $B$, where $B$ is bounded by $K_A / \varepsilon$ and has rank $r$. Requiring an integer matrix to be exactly low-rank can be restrictive and unnatural in practice. Moreover, exact recovery may not be necessary, as a refined entrywise error bound would be sufficient for many applications, particularly under noisy observation settings.

---

> ### Author Rebuttal · Authors · 2025-07-30
>
> We thank you for your comments and the positive evaluation of our paper. We agree with your assessment of the shortcomings of the paper, and we would like to point out that we will try to tackle the case where the underlying matrix is only approximately low-rank in a future paper.
> We also agree with your suggestion of including experiments on real datasets. We will certainly include experiments, paralleling those in other works in the final version if you deem it necessary.
>
> Regarding your questions, we would like to offer our responses below.
>
> 1. Thank you for bringing to our attention these results, as we were not initially aware of them. We have examined the results by Mao et al. and Fan et al. carefully, and we found that their scopes do not cover our use case. We would like to bound $|e_i^TE^kw|$ (this $E$ is $\mathbf{H}$ in Mao et al. and $\mathbf{W}$ in Fan et al.), for all $i\in [n]$ and all $k\le C\log n$ for some large enough constant $C$ like $100$. The probability for the bound to hold must be strong enough to beat the union bounds over all $i\in [n]$ and $k\le C\log n$. This means the probability should be at least $1 - o(n^{-1}\log^{-1}n)$. If we use Lemma 5.4 in Mao et al. for this purpose, we need to choose some $\xi > 1$, and let $t = C\log n$. The factor next to $||v||\_\infty$ in the bound then becomes $(\log n)^{c \log n}$, which grows too fast for our use (our Lemmas D.3 and D.4 has this factor capped at $\log^3 n$). Likewise, Lemmas 4 and 5 in Fan et al. seem to not be directly applicable, since they assume $k$ is a bounded integer. Moreover, in our use case, which is where $E$ has $0$ on its diagonal, we need to involve $||v||_\infty$ in the bound to make it efficient. If we can apply Lemma 4, then Lemma 5 here to arrive at a bound, it will just be $O(\alpha_n^k)$, or $O(n^{k/2})$. This is not much better than the trivial argument $|e_i^TE^kv| \le ||E||^k = O((2n)^{k/2})$ (for our use case only, as theirs also cover the case of non-zero diagonals). We admit that we had not have time to read and analyze their proofs, and it is possible that their techniques can be extended to cover our use case, or help us improve our result even more. We will continue to read their proofs and include a discussion about these results in the final version of the paper.
>
> 2. Thank you for pointing out the typo. Indeed we intended to use $\varsigma$ instead $\kappa$ there. Choosing the names of variables has been challenging for us since this paper uses so many, and we want to keep variable names close to their traditional names in literature if possible. The naming clash for the name $\sigma$ between the singular values and the variance of a random variables has been a particular issue. We will certainly fix it in the final version.

---

> > ### Comment · Reviewer_AT2W · 2025-08-06
> >
> > Thank you for your response. If time permits, I hope the authors will consider adding a brief comparison of their bounds with existing results in the final version. I have no further questions.

---

### Official Review · Reviewer_q3L5 · 2025-07-01

**Clarity:** 3
**Significance:** 3
**Originality:** 3
**Rating:** 5
**Confidence:** 4

**Summary:**

This article looks into the matrix recovery/completion problem, focusing on the assumptions typically utilized to ensure exact recovery is possible. The article identifies three basic assumptions including low rank, incoherence, and sampling density as the necessary assumptions for exact recovery. It then looks into two extra spectral assumptions used in recent works on the small condition number and the large gaps between consecutive singular values, which seem to contradict with each other. It turns out that neither is in fact necessary.
From this the paper establishes the main contribution (Theorem 2.3) that given a sufficiently large spectral norm of the matrix and sufficient sampling rate, the matrix can be exactly recovered with a high probability with a small error. The approximation algorithm is correspondingly based on a truncated SVD. Additionally, in Theorem 3.1, the paper introduces an upper bound on the $l_\infty$-norm of errors in recovery, with a new approach to proof using contour integrals.

**Questions:**

A couple questions:

(1) The two spectral assumptions addressed by the paper in Section 1.3 are that the condition number is small, and the gaps between consecutive singular values are large. My impression is that these statements are rather hard to quantitatively verify. The example provided in the article claims that $\kappa=10.45$ is a large condition number, with which I am not sure I can agree since typically we would call it large when it is in the range of $10^3$. Similarly, in Algorithm 2.2, the largest index $s$ is found by looking for the last 'big' drop in singular values, which is in some way similar to the second spectral statement. I wonder whether the authors can clarify on the refutation of these two assumptions.

(2) The recovery algorithm uses a truncated SVD with an additional scheme to ascertain the cutoff $s$. Since truncated SVD for matrix completion has been widely applied, I am curious if the authors can provide more insight on how the thresholding rule is established.

**Ethical Concerns:**

["NO or VERY MINOR ethics concerns only"]

**Final Justification:**

The concerns from the reviewer have been properly addressed so I have adjusted my rating accordingly.

**Limitations:**

Not limitations but a few minor points:

(1) I wonder whether the authors have considered applying the proposed completion algorithm to a real-world application to demonstrate its performance.

(2) The article denotes the fixed precision of the matrix by $\epsilon$, however, later in the paper it is referred to as the distortion level. Although both are probably supposed to refer to the same entity, I fear this causes some confusion.

(3) In Section 2,1, it is claimed the algorithm is efficient consisting a truncated SVD on a sparse matrix. I am not seeing the sparsity being properly defined so I wonder whether this point can be clarified.

**Quality:**

3

**Strengths And Weaknesses:**

The target problem of the article has long been of interest, including the algorithm to recovery/complete a noisy and partially observed matrix, and theoretical results on the sampling rate to achieve accurate recovery. The literature review on the seminar works and more recent works is quite extensive, including the discussion on the necessity of the basic assumptions and the extra spectral ones. The key contributions include the low-rank approximation algorithm with a proposed method to find the cutoff, and a lower bound on the sampling rate, which are quite significant.

In terms of weakness, my opinion is that I am hoping to see more direct comparison of the sampling rate between the theoretical results proposed with that of the earlier works on matrix completion for their similarity and difference. For example, the results in ''Matrix Completion With Noise'' and ''Exact Low-rank Matrix Completion via Convex Optimization''. I believe that will better highlight the significance of the results by this article.

---

> ### Author Rebuttal · Authors · 2025-07-30
>
> We thank you for your comments and the positive evaluation of our paper. We would like to address the following concerns and questions of your review:
>
> **1. Weaknesses of the paper:**
>
> 1.  We acknowledge that we did not set aside a separate section to summarize and clarify the advantages and possible disadvantages of our results versus the other works. In an early version of the paper, we had a comparison table, which was cut from the final draft after we restructured the paper and removed the discussions and comparisons with several other papers that we felt not as relevant as the ones we kept. We did this to comply with the page limit. If you prefer, we will try to bring it back, with appropriate adjustments, probably in Section 3.3.
>
> 2. Let us respond specifically to clarify your concern about the lack of direct comparisons with "Matrix Completion with Noise" by Candes and Plan (reference [8] in our paper) and "Exact Low-rank Matrix Completion via Convex Optimization" by Candes and Recht (reference [1]).
>
>   - The result of [1] is one of most influential works in Matrix Completion that deal with the pure (noiseless) problem. It was followed shortly by the papers of Candes and Tao (reference [2]) and Recht (reference [3]) that direct improved on it. Therefore, it suffices to compare our result with [3]. Ignoring the factors dependent on $\mu_0$ (the coherence) and $r$ (the rank of the original matrix), [3] has achieved near-optimal sampling rate, being off by a $\log(m + n)$ factor from the theoretical limit $(m + n)\log(m + n)$. Our result has a sampling rate off by $\log^9(m + n)$, but otherwise is optimal. Its main advatage is the ability to deal with random, independent noise, which is not addressed in [1, 2, 3]. Our algorithm is also simpler and faster in practice (it basically computes a low rank approximation of the input).
>
>   - The result (Theorem 7) in [8] is more general than ours in the sense thay it holds for any type of noise, whereas ours only apply to noises that are independent among entries and universally bounded by a known upper bound. We also need the extra assumption that $||A||$ is large enough (which we have argued is reasonable and perhaps necessary for exact recovery in Remark 2.4). However, their method only achieves recovery in the Frobenius norm, or RMSE, sense, while ours gives recovery in the infinity norm sense, which becomes exact after rounding off. Their method, being convex optimization, runs in time $O(|\Omega|^2(m + n)^2)$, which is at least $O((m + n)^4\log^2(m + n))$ for recovery to be theoretically possible (see the survey [15] in our paper). On the other hand, our method runs in time $O(r|\Omega| + mn)$, being simply a truncated SVD algorithm, making it much faster than [8]. Another point is the effectiveness of the methods. To achieve a RMSE error (which is $||\hat{A} - A||_F / \sqrt{mn}$) within some $\varepsilon$, the method in [8] needs $p \ge \epsilon^{-2} ||Z||_F^2 / (m + n)$, which is not possible even in the simple cases like $Z$ having independent entries in $\{-1, 0, 1\}$ with uniform distribution (setting $p = 1$, which means full observation, is still not enough if $\varepsilon = 0.1$). Our method achieves $||\hat{A} - A||\_\infty \le \varepsilon$, which is stronger, while only needing $m$ and $n$ to be large enough and $p \ge C(r, \mu_0)\max\{\log^4 (m + n), \varepsilon^{-2}\} \log^6(m + n)$, which is much lower and achievable for the aforementioned simple noise model.
>
> 3. We also want to offer a brief comparison with the works of Keshavan, Oh and Montanari (references [6] and [7]) that deal with both the noiseless and noisy cases. Our sampling rate is slightly higher than theirs, being off by a few $\log (m + n)$ factors, and we need the extra assumption that $||A||$ is large enough (see the previous item and Remark 2.4 in our paper). On the other hand, we remove the dependence on the condition number that they have, and our method achieves an infinity norm recovery while theirs stop at RMSE recovery.
>
> **2. Questions.**
>
> 1. Although  a condition number $\kappa \approx 10.45$ may be small, it can get very large if raised to high powers, making bounds inefficient. For example, the papers [6] and [9] that we discuss throughout Section 1 involve $\kappa^6$ and $\kappa^4$ in their sampling rate bounds. Our result avoids this growth factor by removing $\kappa$ completely.
>
> 2. A high-level explanation for our thresholding rule is as follow. We view the observed matrix, rescaled by $p^{-1}$, as a perturbed version of the original matrix, namely $p^{-1}A_{\Omega, Z} = \tilde{A} = A + E$, where $E$ is a random matrix of mean $0$ and independent entries. One can view $E$ as a type of "noise" that includes both the noise $Z$ and the noise caused by the random sampling. We simply want to cut off the singular values of $\tilde{A}$ at a level above $||E||$, as the BBP phenomenon shows that the part of $A$ below that is fully absorbed by $E$ and becomes indistinguishable from noise. Under the assumptions we make, the signal extracted from the spectrum of $\tilde{A}$ above this threshold is close enough to $A$ to allow us to recover it. We use the same argument in Remark 2.4 to show the necessity of the lower bound on $\sigma_1 = ||A||$.
>
> **3. Limitations:**
>
> 1. Thank you for the suggestion of including experiments on real datasets. As pointed out by other reviews, our assumption of low rank still does not fully reflect real-life datasets, which are often only approximate low-rank, so initially we wanted to focus on the fact our result breaks the two theoretical barriers involving the condition number and large singular value gaps as the main highlight. We will certainly include experiments, paralleling those in other works in the final version if you deem it necessary. We are also working on an extension of this paper which would cover the case when the truth matrix is only approximately low rank.
>
> 2. We apologize for the potential confusion caused by the word "distortion level". We can revert it back to "discretization unit", for example.
>
> 3. We use the word "sparse" simply to point out that, in the worst case that our method can still guarantee recovery with high probability, we allow the sampling rate $p$ (the density of non-zero entries in the observed matrix $A_{\Omega, Z}$) to be very small, being at most $C(\mu_0, r) (m^{-1} + n^{-1}) \log^C(m + n)$. We acknowledge that $A_{\Omega, Z}$ can become dense if $p$ increases. This will increase the time our SVD step takes, but also makes it easier to recovery $A$. As mentioned above, the time complexity $O(r|\Omega| + mn) = O((pr + 1)mn)$ is still substantially better than nuclear norminimzation for any value of $p$.

---

> > ### Comment · Reviewer_q3L5 · 2025-08-05
> >
> > I thank the authors for their detailed explanation to my queries, including the comparison to the results in the seminar papers which I particularly appreciate. The authors' statement towards my other questions are satisfactory as well.
> >
> > This is good work.

---

### Official Review · Reviewer_KmFY · 2025-07-05

**Clarity:** 3
**Significance:** 3
**Originality:** 3
**Rating:** 5
**Confidence:** 4

**Summary:**

This paper studies exact matrix completion in a setting in which the underlying matrix has rank r and  entries that are multiple of a known fixed number epsilon. The authors show that exact reconstruction is possible (in the case of fixed rank) as soon as the number of observed entries is larger than $(m+n)*log(m+n)^4$ which is nearly optimal.
The important point is that no assumption on the condition number or separation between singular values is made.

**Questions:**

I do not have questions.

**Ethical Concerns:**

["NO or VERY MINOR ethics concerns only"]

**Limitations:**

Yes

**Quality:**

3

**Strengths And Weaknesses:**

Strenghts:
1. Removing the dependence on the condition number and separation between singular values is significant.
2. The proof technique is interesting.

Weaknesses:
1. Several key ideas are actually based on Tran Vu (2024)
2. The algorithm is a small variation over method studied before.
3. The assumption of discrete entries together with small rank is unrealistic. For instance the netflix ratings matrix is discrete but does not have small rank.

---

> ### Author Rebuttal · Authors · 2025-07-30
>
> We thank you for your comments and the positive evaluation of our paper. We agree with your assessment of the shortcomings of the paper, and we would like to point out that we will try to tackle the case where the underlying matrix is only approximately low-rank in a future paper. As a matter of fact, the contour method used in the paper of Tran and Vu (that we follow)  applies for the approximately low rank case, when the matrix has few large leading eigenvalues and the rest are small.  However, to prove perturbation bound in the infinity norm as in this paper, this method needs to be modified significantly. We are working on the details at this moment, but are pretty positve on the outcomes.

---

### Decision · Program_Chairs · 2025-09-17

**Decision:**

Accept (poster)

**Comment:**

One reviewer was still not satisfied with the paper, arguing that a ground truth which is low rank and has integer entries is not so realistic. The other reviewers also acknowledged this but for them this concern was outweighed by the nice technical contributions of the paper, and removing assumptions in prior work on condition number and singular value gaps that were needed even in this case. I tend to agree with the majority here, and think the ideas here are significant and this will be impactful for future work. Thus, I recommend acceptance.